# Longitudinal transmission of bacterial and fungal communities from seed to seed in rice

Hyun Kim [1], Jongbum Jeon[2,6], Kiseok Kieth Lee[1,7] & Yong-Hwan Lee [1,2,3,4,5✉]

Vertical transmission of microbes is crucial for the persistence of host-associated microbial communities. Although vertical transmission of seed microbes has been reported from diverse plants, ecological mechanisms and dynamics of microbial communities from parent to progeny remain scarce. Here we reveal the veiled ecological mechanism governing transmission of bacterial and fungal communities in rice across two consecutive seasons. We identify 29 bacterial and 34 fungal members transmitted across generations. Abundance-based regression models allow to classify colonization types of the microbes. We find that they are late colonizers dominating each community at the ripening stage. Ecological models further show that the observed temporal colonization patterns are affected by niche change and neutrality. Source-sink modeling reveals that parental seeds and stem endosphere are major origins of progeny seed microbial communities. This study gives empirical evidence for ecological mechanism and dynamics of bacterial and fungal communities as an ecological continuum during seed-to-seed transmission.

[1] Department of Agricultural Biotechnology, Seoul National University, Seoul 08826, Republic of Korea. [2] Interdisciplinary Program in Agricultural Genomics, Seoul National University, Seoul 08826, Republic of Korea. [3] Center for Plant Microbiome Research, Seoul National University, Seoul 08826, Republic of Korea. [4] Plant Immunity Research Center, Seoul National University, Seoul 08826, Republic of Korea. [5] Research Institute of Agriculture and Life Sciences, Seoul National University, Seoul 08826, Republic of Korea. [6] Present address: Korea Bioinformation Center, Korea Research Institute of Bioscience and Biotechnology, Daejeon 34141, Republic of Korea. [7] Present address: Department of Ecology and Evolution, The University of Chicago, 1101 East 57th Street, Chicago, IL 60637, USA. ✉email: yonglee@snu.ac.kr

Human, animals, and plants are not sterile entities but biological unions associated with microbial communities. Microbial communities can support plant physiology[1], development[2,3], and immunity against abiotic and biotic stresses[4,5], ultimately increasing host fitness. Host-associated microbial community consists of vertically transmitted (transmitted from parents) and horizontally transmitted (acquired from environments) fractions[6]. A comparative study reported that host fitness significantly reduces when vertically transmitted microbes are eliminated[7]. Thus, the understanding on vertical transmission of microbial community is crucial for unveiling reciprocal relationship between host and microbial community. In plant, seed contributes to the transmission of microbial community from generation to generation.

Seed-to-seed transmission of a bacterial or fungal community across generations has been reported in *Setaria viridis*[8], *Crotalaria pumila*[9], *Cucurbita pepo*[10,11], radish[12], potato[13], and tomato[14]. These studies mainly showed vertically transmitted bacteria or fungi detected across two or three generations, a potential source of a seed microbial community at a particular developmental stage of a plant host, and assembly mechanisms affecting a seed microbial community. Although these studies gave the ecological understanding on the vertical transmission of seed bacterial and fungal communities, there are still open questions. First, little is known about how seed bacterial and fungal communities are assembled during seed-to-seed transmission at the temporal scale, such as a life cycle of plants. Second, it is not clear when seed bacterial and fungal communities are assembled. Third, knowledge on which sources where seed microbes originate, such as soils and plant endosphere, contribute to the establishment of seed microbial communities along with plant development remains scarce.

Assembly and dynamics of host-associated microbial community are affected by niche and neutral process. Niche process hypothesizes that deterministic factors including species traits, interspecies interactions, and environmental filtering govern community structure, whereas neutral process assumes that community structures are assembled by stochastic processes of birth, death, colonization, extinction, and speciation[15]. In host-associated microbial communities, host-related factors are generally considered as deterministic factors. Host immune system and functional genes regulate the colonization of specific bacterial taxa[16–18]. In humans and primates, heritability concept, a measure of genetic influence on variation in phenotypic traits, has been adopted. This approach helps find microbes and community-associated traits that are under host control by linking variation in microbial community to host genetic variation[19,20]. Neutral community models proposed by Hubbell[21] and Sloan et al.[22] have been widely used for investigating contribution of neutral process to community assembly. This approach showed that host-associated microbial communities are also significantly affected by neutral process[23,24]. Although host developmental stage significantly contributes to the assembly of microbial communities[25–30], it is not clear whether the temporal effect is deterministic, neutral, or both. A previous study reported that ecological drift governs the assembly of radish seed bacterial and fungal communities using the regional species pool and local community concept[12]. However, the ecological forces governing the longitudinal dynamics of bacterial and fungal communities from developing to developed seeds and taxa affected by the ecological forces have not been elucidated yet.

Microbial associations are also important for understanding the assembly of microbial communities. Network inference is widely applied to investigate association patterns in microbial communities in diverse environments. Although network inference has limitations resulting from a correlation-based prediction tool, it has provided information on unprecedent complexity, interactive dynamics, and potential interactions of prokaryotic and eukaryotic organisms[31–33]. Meta-networks (global networks or metacommunity-scale networks) can provide a profitable resource for understanding microbial associations patterns and interlinkage of local communities at the geographic level[32,34]. Thus, meta-network will be useful for examining dynamics of microbial associations at not only spatial scale but also temporal scale.

Here, we addressed the veiled parts in the seed-to-seed transmission of bacterial and fungal communities by examining bacterial and fungal communities associated with field-grown rice plants and soil environments during the life cycle of rice. We investigated longitudinal dynamics of rice-associated microbial communities from seeds, leaves, stems, roots, rhizosphere, and bulk soil during rice development across two consecutive years. The longitudinal sampling and comprehensive statistical analyses allowed us to reveal the longitudinal transmission, associations, and sources of seed bacterial and fungal communities. We also found that the longitudinal transmission of seed microbial communities is governed by both host and neutral factors by adopting heritability concept and neutral theory.

## Results

**Experimental design and sequencing results**. In the previous study, we proposed that microbes are vertically transferred in seeds[35]. To elucidate the possibility of constitutive vertical transmission of bacterial and fungal communities during rice growth, we collected developing and developed rice seeds in two consecutive years from three fields located in two geographically distant locations (2017 and 2018). Other plant compartments (leaves, stems, and roots) and corresponding soil samples were parallelly collected to examine whether bacterial and fungal communities in soils and plant compartments act as microbial pools (or sources) of seed microbial communities (Supplementary Fig. 1). In each sampling point, plants and three soil samples were obtained as biological repeats. As a result, we collected 30 rice plants (9 plants for 2017; 21 plants for 2018) and 78 soils (36 bulk soils for 2017; 21 bulk and 21 rhizosphere soils for 2018) during rice development. The collected leaves and stems were further divided into subgroups according to their positions (Supplementary Fig. 1). A total of 1164 rice and soil samples including seeds, leaves, stems, roots, rhizosphere, and bulk soils were used for this study. Details on sampling strategy and sample preparation are available in the Method section. Bacterial and fungal communities were investigated with V4 region of 16S ribosomal RNA (rRNA) genes and internal transcribed spacer (ITS) regions of nuclear rRNA genes, respectively. After discarding chimeras, plant-derived reads (mitochondria, plastid, and rice ITS sequences), and archaeal reads, 15,975 bacterial (seed, 2,417,126 reads; leaf, 1,282,010 reads; stem, 3,613,146 reads; root, 2,140,375 reads; rhizosphere, 1,069,424 reads; bulk soil, 1,291,831 reads) and 5963 fungal (seed, 4,654,040 reads; leaf, 10,133,775 reads; stem, 13,916,068 reads; root, 2,942,672 reads; rhizosphere, 1,582,409 reads; bulk soil, 1,884,782 reads) operational taxonomic units (OTUs) were used for downstream analyses.

**Temporal and spatial shifts in rice-associated bacterial and fungal communities**. When taxonomic affiliations of the bacterial and fungal communities were examined, the dominance of phyla or classes varied with plant compartments (Fig. 1a; Supplementary Figs. 2–4). In the bacterial community, Alphaproteobacteria (orders Rhizobiales, 3.8–41%; Sphingomonadales, 4–24.3%) and Gammaproteobacteria (orders Enterobacteriales, 0.1–60.1%; Betaproteobacteriales. 1.7–36.4%) were abundant in plant endosphere (Fig. 1a; Supplementary Fig. 2; Supplementary Data 1).

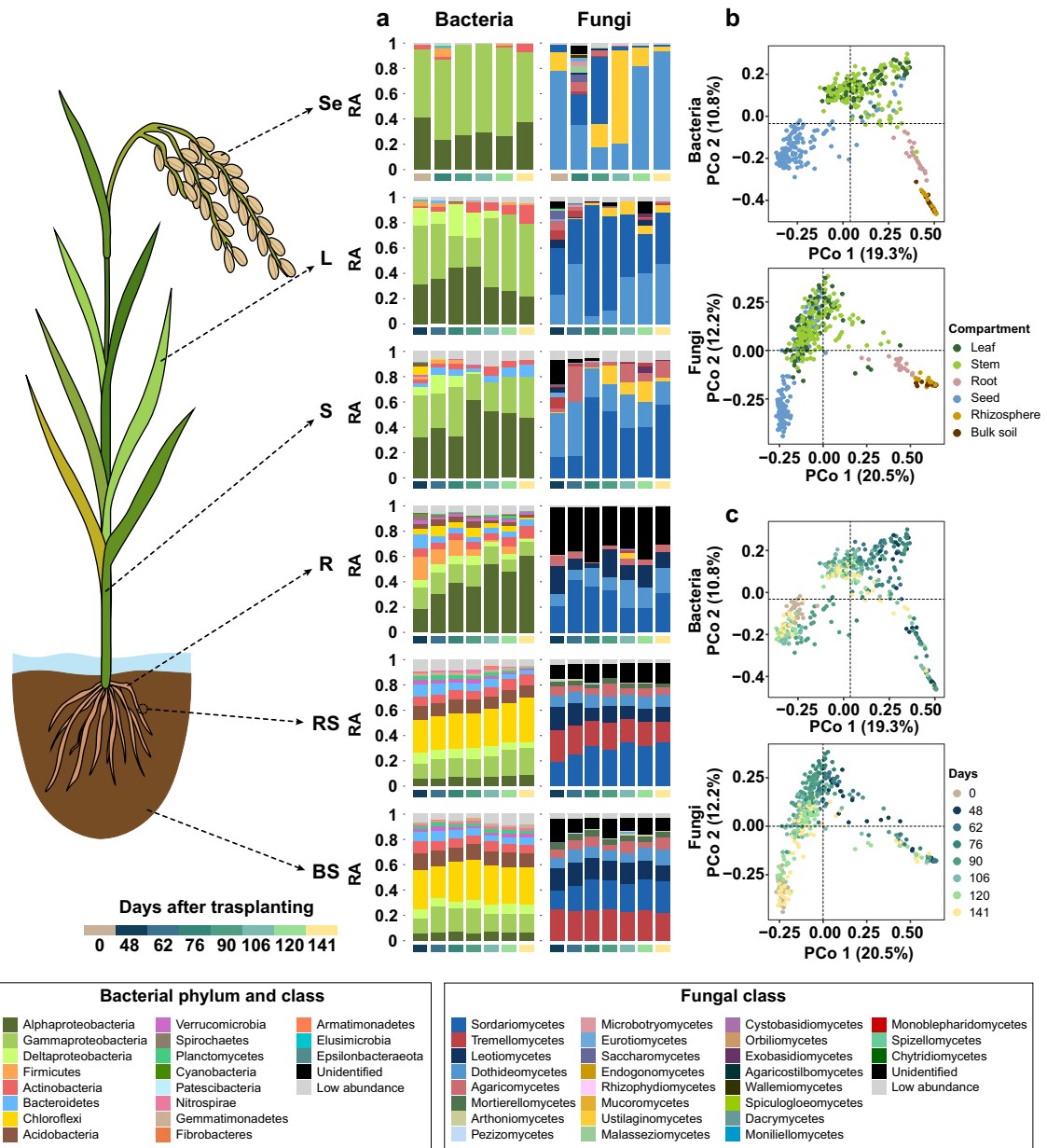

**Fig. 1 Compositional variations in rice-associated bacterial and fungal communities over time. a** Distribution of relative abundances of bacterial and fungal communities in different compartments during the growing season of rice. In seeds, each bar corresponds to the mean relative abundance of 27 biological replicates of each age (days after transplanting) except seeds collected at 76 days after transplanting (18 biological replicates). In other compartments, each bar represents the mean relative abundance of 9 replicates (3 biological replicates × 3 technical replicates) of a particular age. Details on replicates are available in Methods. The relative abundances of each replicate of a particular age are available in source data in the Supplementary Data 1 and Supplementary Figs. 2, 3 in the Supplementary Information. The bars are ordered by the rice age as indicated by the colored bars under each bar. BS, bulk soil; RS, rhizosphere soil; R, root endosphere; S, stem endosphere; L, leaf endosphere; Se, seeds; RA, relative abundance. **b** Compositional variations of bacterial and fungal communities in 2018 samples. Compositional variations among samples were estimated via principal coordinates analysis (PCoA) based on Bray-Curtis distance metric. Bray-Curtis distance was calculated from the mean abundance tables of technical replicates. The samples are colored by compartments. **c** The same plot of bacterial and fungal communities colored by days after transplanting. Dots represent biological replicates of each rice compartment-associated bacterial or fungal community.

In the bulk soils and rhizosphere, Chloroflexi (class Anaerolineae, 21.5–26.9%; order Anaerolineales, 11.7–14.9%) was dominant compared to plant endosphere (Fig. 1a; Supplementary Fig. 2; Supplementary Data 1). This compositional difference was also found in the fungal community. In the plant endosphere, Sordariomycetes (orders Hypocreales, 0.1–38.2%; Magnaporthales, 0.1–11.9%), Dothideomycetes (order Pleosporales 8.9–91.6%), Ustilaginomycetes (order Ustilaginales, 0.21–78.9%) were abundant across years and locations (Fig. 1a; Supplementary Figs. 3, 4),

whereas Tremellomycetes (order Cystofilobasidiales, 11.9–19.5%), Agaricomycetes (order Agaricales, 2.3–5.7%), Sordariomycetes (order Sordariales, 9.3–27.2%), and Leotiomycetes (orders Helotiales, 6.1–12.2%; Thelebolales, 3.6–8.8%) dominated the rhizosphere and bulk soil fungal communities (Fig. 1a; Supplementary Fig. 3; Supplementary Data 1). The occupancy ratio of bacterial and fungal taxa also varied with rice ages (sampling timepoints). For example, in root endosphere, relative abundance of Alphaproteobacteria increased from 18% to 60% along with the rice age

(Fig. 1a). Ustilaginomycetes belonging to Basidiomycota was abundant in seeds (67% for 106 days after transplanting and 11.2% for 120 days after transplanting) and stems adjacent to seeds (S6 [stem at 50–60 cm of height], 11–48%; S7 [stem at 60–70 cm of height], 10–52%; S8 [stem at 70–80 cm of height], 9–48%; S9 [stem at 80–90 cm of height], 9–43%) (Supplementary Fig. 3; Supplementary Data 1). The observed compositional variations by compartment and age were supported by ordination analyses and permutational analysis of variance (PERMANOVA). In both 2017 and 2018 samples, rice compartments contributed the most to the community variations (Fig. 1b; Supplementary Fig. 5). When considered all samples across 2017 and 2018, both bacterial and fungal communities were clearly clustered by rice compartments (Supplementary Fig. 6). This is supported by PERMANOVA (Supplementary Data 1; Bacteria: $R^2 = 0.232$, $P = 0.0001$ [2017], $R^2 = 0.266$, $P = 0.0001$ [2018]; Fungi: $R^2 = 0.297$, $P = 0.0001$ [2017], $R^2 = 0.283$, $P = 0.0001$ [2018]). The effect of compartments was followed by host age (Fig. 1c; Supplementary Fig. 5). PERMANOVA showed that 9.83% (2017) and 10.88% (2018) of bacterial variance and 12.03% (2017) and 13.96% (2018) of fungal variance could be explained by rice age (Supplementary Data 1).

**Vertically transmitted OTUs dominate seed microbial communities during seed maturation.** To investigate the vertical transmission of bacterial and fungal communities, we hypothesized that community dissimilarity between parent and progeny seeds decreases when microbes are vertically transmitted. Pairwise Jaccard (based on presence/absence patterns) and Bray-Curtis (based on abundance patterns) community dissimilarity between parent (Se0, seeds at the sowing stage) and progeny seeds showed that bacterial and fungal compositions of progeny seeds become similar to those of parent seeds (Fig. 1b, c; Supplementary Fig. 7). These results demonstrated that abundance and presence/absence patterns of progeny seed microbial communities become similar to parent seeds, suggesting that membership of seed bacterial and fungal communities is transmitted from generation to generation.

To dig into the vertical transmission of bacterial and fungal communities during rice development, we defined transmitted OTUs which were commonly distributed in seeds harvested in October 2017, seeds sowed in May 2018, and progeny seeds harvested in October 2018 from the average over plant replicates. The remaining OTUs of seed bacterial and fungal communities were considered as transient OTUs that could colonize in seeds but not persist in three conditions. Bacterial transmitted OTUs consisted of *Pantoea*, *Methylobacterium*, *Sphingomonas*, *Pseudomonas*, and *Allorhizobium-Neorhizobium-Pararhizobium-Rhizobium* (Supplementary Data 2). In the fungal community, *Nigrospora*, *Moesziomyces*, and unidentified genus belong to the order Pleosporales, *Curvularia*, and *Pyrenochaetopsis*. Transmitted OTUs accounted for 20% (29 out of 142 OTUs) and 15% (34 out of 223 OTUs) of total seed OTUs in the bacterial and fungal communities, respectively. Although number of transmitted OTUs was lower than that of transient OTUs, transmitted OTUs accounted for above 80% of relative abundances of the bacterial and fungal communities from mature seeds. The relative abundance of transmitted OTUs was dramatically changed during seed maturation (Fig. 2). Numbers and relative abundances of bacterial transmitted OTUs were lowest at the early developmental stage of seeds. During heading and ripening stages, relative abundance of bacterial transmitted OTUs dramatically increased (Fig. 2a). Similar temporal patterns were observed in the fungal community, but fungal transmitted OTUs dominated the seed communities from the early developmental stage (Fig. 2a). Since total relative abundance of transmitted

OTUs was recovered during seed maturation, we next checked whether the relative abundance of individual transmitted OTUs as well as membership is recovered during seed maturation. We compared transmitted microbial community composition, which consists of 29 bacterial and 34 fungal OTUs, of parent and developing progeny seeds using Bray–Curtis dissimilarity index to consider the pairwise differences in OTU abundances. In the bacterial and fungal communities, community dissimilarity values significantly decreased following seed maturation (Fig. 2b). This result indicates that the abundances of transmitted OTUs in the progeny seeds become similar to that in the parent seeds. This result was further supported by predicting successional modes of seed bacterial and fungal OTUs. We identified three successional modes of microbial OTUs (early successor, mid-successor/no trend, and late successor) by performing a linear regression between host age and relative abundance of OTUs. Most of the transmitted OTUs (24 out of 29 bacterial OTUs; 26 out of 34 OTUs) were predicted as late successors that dominated seed microbial communities at the late seed maturation stage (Fig. 2c; Supplementary Fig. 8; Supplementary Data 3). Meanwhile, non-transmitted OTUs belonged to early successors (Supplementary Fig. 8). These results suggest that colonization of transmitted OTUs may be affected by host physiology or environmental filtering during seed maturation.

**Neutrality affects the succession of seed microbial community.** To find whether the assembly of vertically transmitted OTUs is affected by temporal changes in niche environment, we adopted a heritability concept. Heritability is a measure of genetic effects on variation in phenotypes or traits. In the present study, we considered niche environment as an analog of genetic influence that affects plant phenotypes. The composition of microbial communities was also considered as one of the plant phenotypes. This approach allowed us to evaluate the effects of temporal changes in niche environment on abundances of transmitted OTUs during seed maturation. For this analysis, OTUs which relative abundance is lower than 0.0001 were removed from OTU abundance table. Niche responsiveness of 28 out of 29 bacterial transmitted OTUs and 24 out of 34 fungal transmitted OTUs was estimated. Niche responsiveness of bacterial transmitted OTUs ranged from 0.0191 to 0.9523, whereas that of fungal transmitted OTUs ranged from 0.0125 to 0.9333 (Fig. 3a). A total of 21 transmitted OTUs (13 bacterial and 8 fungal OTUs) showed high level of niche responsiveness ($h^2 > 0.4$). Two bacterial OTUs showed moderate niche responsiveness ($0.2 < h^2 \leq 0.4$). Niche responsiveness of other 30 transmitted OTUs (14 bacterial and 16 fungal OTUs) was low ($h^2 \leq 0.2$). These results suggest that vertically transmitted OTUs are not equally affected by host physiology or environmental filtering during seed maturation.

To investigate the effect of neutrality on the assembly of vertically transmitted OTUs in seeds, occurrence–abundance relationship was examined using Sloan's neutral model (Fig. 3b). Among 29 bacterial transmitted OTUs, seven OTUs belonging to *Pantoea*, *Methylobacterium*, *Sphingomonas*, and *Pseudomonas* were predicted to be assembled by deterministic process (Supplementary Data 4). The other OTUs were affected by neutrality (Supplementary Data 4). In the fungal community, meanwhile, 22 out of 34 transmitted OTUs belonging to *Cladosporium*, *Moesziomyces*, *Alternaria*, *Curvularia*, and *Nigrospora* were affected by deterministic process (Fig. 3b; Supplementary Data 4). These results suggest that vertically transmitted fungi are mainly assembled in a niche-based way, whereas vertically transmitted bacteria are mostly affected by neutrality. Using heritability (niche responsiveness) concept and Sloan's neutral model, we also found that six bacterial OTUs belonging

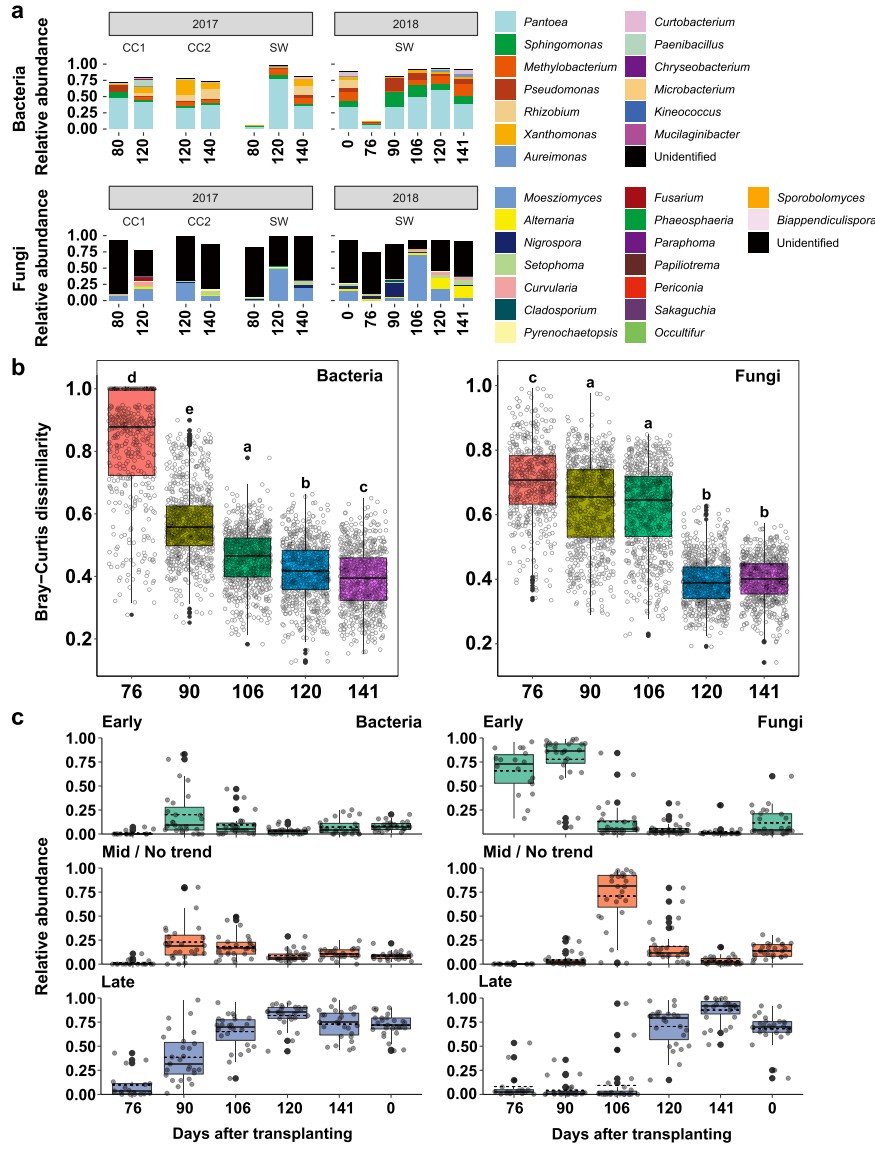

**Fig. 2 Distribution and successional changes of vertically transmitted bacterial and fungal communities. a** Relative abundance of vertically transmitted bacterial and fungal communities during seed development. Mean relative abundances of bacterial and fungal OTUs were estimated at the genus level. Exact values of relative abundances and standard deviations are available in Supplementary Data 2. Bar plots are depicted according to sampling year and geographic sites. The numbers in x-axis indicate sampling point (days after transplanting to fields). CC1, Chuncheon 1; CC2, Chuncheon 2; SW, Suwon. **b** Pairwise dissimilarity between vertically transmitted bacterial and fungal communities in parent seeds (Se0) and progeny seeds collected from 76 to 141 days after transplanting. Pairwise community dissimilarity was calculated using Bray-Curtis dissimilarity index. Boxes and lines in the boxes represent inter-quantile range (Q3–Q1) and median of dissimilarity values, respectively. Black-filled dots indicate potential outliers. Lower and upper whiskers show minimum and maximum values of community dissimilarity in each comparison pair. Statistical significance of each compartment by rice age was estimated using Kruskal-Wallis test with two-sided Dunn's test since dissimilarity values did not meet the normal distribution (Shapiro-Wilk normality test, $P < 2.2e$-16 for bacteria and fungi). The letters indicate statistical significance ($P < 0.05$). The statistical results are available in the source data in Supplementary Data 2. **c** Successional changes of cumulative relative abundance of vertically transmitted bacterial and fungal communities. Succession modes were classified based on linear regression between time and relative abundance of each OTU (Early, $t < 0$, $P < 0.05$; Mid/no trend, $P > 0.05$; Late, $t > 0$, $P < 0.05$). The numbers in x-axis indicate sampling point (days after transplanting to fields). The number 0 indicates parent seeds. The other numbers indicate progeny seeds. Boxes and lines in the boxes represent inter-quantile range (Q3–Q1) and median of cumulative relative abundances of OTUs in the succession groups in each time, respectively. Black dashed lines in the boxes indicate the mean cumulative relative abundances. Black-filled and gray dots indicate potential outliers and the cumulative relative abundances in each biological replicate ($n = 18$ for 76 days after transplanting; $n = 27$ for 90–141 days after transplanting), respectively. Lower and upper whiskers show minimum and maximum relative abundance values.

to *Pantoea*, *Methylobacterium*, *Sphingomonas*, and *Pseudomonas* (B1, B2, B3, B4, B5, and B35) and five fungal OTUs belonging to *Moesziomyces*, *Pyrenochaetopsis*, order Pleosporales, and family Phaeosphaeriaceae (F3, F4, F18, F54, and F85) were commonly predicted to be assembled by niche process in which temporal changes in niche environments may act. All OTUs were predicted

as vertically transmitted OTUs, suggesting that rice may transfer these OTUs to progeny seeds actively rather than other OTUs.

**Microbial communities of stem and parental seeds are major sources of those of progeny seeds.** To find the origin of microbial communities in progeny seeds, we investigated source–sink

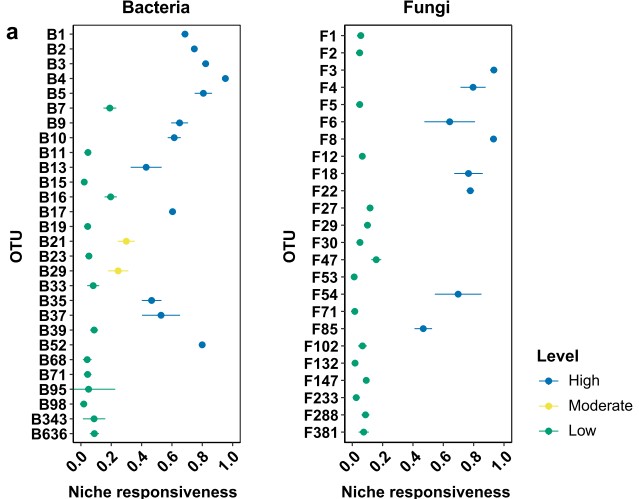

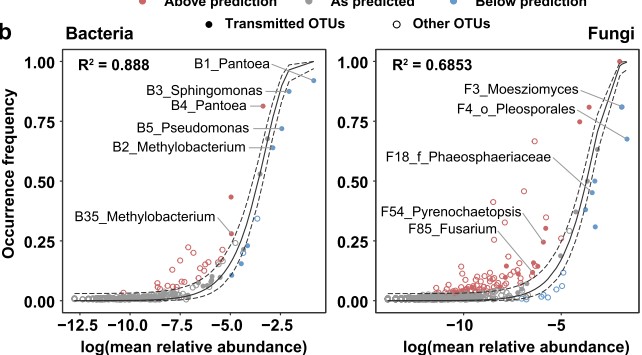

**Fig. 3 Heritability and neutral process in the formation of seed microbial community. a** Heritability estimates for vertically transmitted bacterial and fungal OTUs. Dots and error bars indicate mean heritability estimates and standard deviations from bootstrapped data (number of simulations = 100). Dots are colored by levels of estimates (High, $h^2 > 0.4$; Moderate, $0.2 < h^2 \leq 0.4$; Low, $h^2 \leq 0.2$). **b** Sloan's neutral model for seed bacterial and fungal communities. Each dot indicates OTUs in the bacterial and fungal communities. The predicted neutral model (solid black line) is surrounded by 95% confidence intervals (dashed lines). The frequency of occurrence of bacterial and fungal OTUs during seed development (y-axis) is plotted against OTU's relative abundance (x-axis) to indicate that the OTU is most likely neutrally distributed (gray dots) or selected (red and blue dots). Filled dots correspond to the vertically transmitted bacterial and fungal OTUs, whereas hollow dots denote other seed OTUs. The names of bacterial and fungal OTUs indicate the OTUs commonly predicted to be affected by niche process using heritability (niche responsiveness) analyses and Sloan's neutral models.

relationship between rice compartments and progeny seeds. Since programmed cell death occurs systemically throughout the rice body at the harvest stage, we did not consider microbial communities at the harvest stage as a source. We treated seed samples collected at the harvest stage as sinks, considering seed samples at the earlier timepoints and plant samples (leaves, stems, root, rhizosphere, and bulk soil) as sources. Larger contribution was found in stem endosphere in the bacterial community (40.1%), whereas developing seeds were more contributed as a source in the fungal communities (31.2%) (Fig. 4a). To find temporal contribution of sources to sinks, we estimated contribution of the microbial communities associated with plant and soil compartments at each timepoint. In the bacterial community, parental seeds (mean ± s.d.; 3.25 ± 1.72%) and stem endosphere at the ripening stages (mean ± s.d.; 106, 11.04 ± 1.75%; 120, 18.14 ± 5.31%) served as

sources for progeny seeds (Fig. 4b). Meanwhile, parental seeds (mean ± s.d.; 0, 17.35 ± 6.27%) and seeds at the ripening stages (mean ± s.d.; 106, 2.51 ± 0.95%; 120, 11.28 ± 8.76%) more contributed as sources for fungal communities of progeny seeds than stem endosphere (Fig. 4b). These results suggest that microbial communities in parental seeds and stem endosphere are major sources of progeny seed microbial communities. Based on these results, we further hypothesized that vertically transmitted OTUs prevalently exist in stem endosphere if they are moved through the stem space following the rice development. Most bacterial and fungal transmitted OTUs were revealed as core OTUs (Supplementary Result 1; Supplementary Data 5) in at least one aboveground compartment, and broadly detected in the aboveground compartments (Fig. 4c, d; Supplementary Result 1; Supplementary Fig. 9). These results suggest that vertically transmitted bacterial and fungal OTUs could be systemically and persistently colonized in the aboveground compartment.

**Meta-network reveals commonality and specificity of microbial associations**. To understand temporal and spatial dynamics of microbial associations during rice development and seed-to-seed transmission of bacterial and fungal communities, we next constructed bacterial-fungal co-occurrence networks of different times and spaces. A total of 101 bacterial-fungal co-occurrence networks which correlation coefficients were higher than 0.3 or lower than −0.3 were constructed (Supplementary Fig. 10). The microbial networks of belowground compartments (mean ± s.d.; 587 ± 107 nodes and 3245 ± 944 edges) showed higher complexity than those of aboveground compartments (mean ± s.d.; 229 ± 123 nodes and 1632 ± 1445 edges) (Supplementary Data 6). Although network properties of microbial networks differed with time and space (Supplementary Data 6), we hypothesized that composition of microbial associations is similar among co-occurrence networks of similar niches. To prove this assumption, we constructed a global microbial network consisting of 101 microbial networks. The resulting meta-network consisted of 6515 nodes and 198,748 edges (Fig. 5a). The meta-network consisted of 17 modules (subnetworks), but 94.07% of nodes belonged to top 4 modules (module 1 [25.1%], module 3 [16.04%], module 6 [17.04%], and module 11 [35.89%]). The composition of edges in top 4 modules exhibited the specificity of microbial associations throughout rice compartments. Modules 1 and 3 mainly consisted of edges detected in soil (bulk and rhizosphere soils) and root endosphere, respectively (Fig. 5b). On the other hand, modules 6 and 11 were dominated by edges detected in aboveground compartments (leaf, stem, and seed endospheres) (Fig. 5b). Interestingly, edges found in vegetative (48 and 62 days after transplanting) and early reproductive stages (76 days after transplanting) dominated module 11, whereas edges in late reproductive (90 days after transplanting) and ripening stages (106, 120, and 141 days after transplanting) were abundant in module 6 (Fig. 5b; Supplementary Data 7). These results suggest that microbial networks differ from ecological niches for microbial communities and the transition of microbial associations in the aboveground compartments occurs around the reproductive stage. Commonality and specificity of bacterial-fungal associations were further corroborated by pairwise comparison of edge profiles of 101 bacterial–fungal associations using Jaccard similarity index ($J$) (Fig. 5c). Bacterial-fungal associations in the leaf, stem, and root endosphere at the ripening stages (106, 120, and 140 days after transplanting) were clustered tightly (Fig. 5c; Supplementary Result 2). These similarity patterns among microbial networks was also observed in the similarity of hub profiles identified based on degree centrality (Supplementary Fig. 11; Supplementary Data 8). These results suggest that the systemic changes in the endospheric environments in rice during the ripening stage may lead to

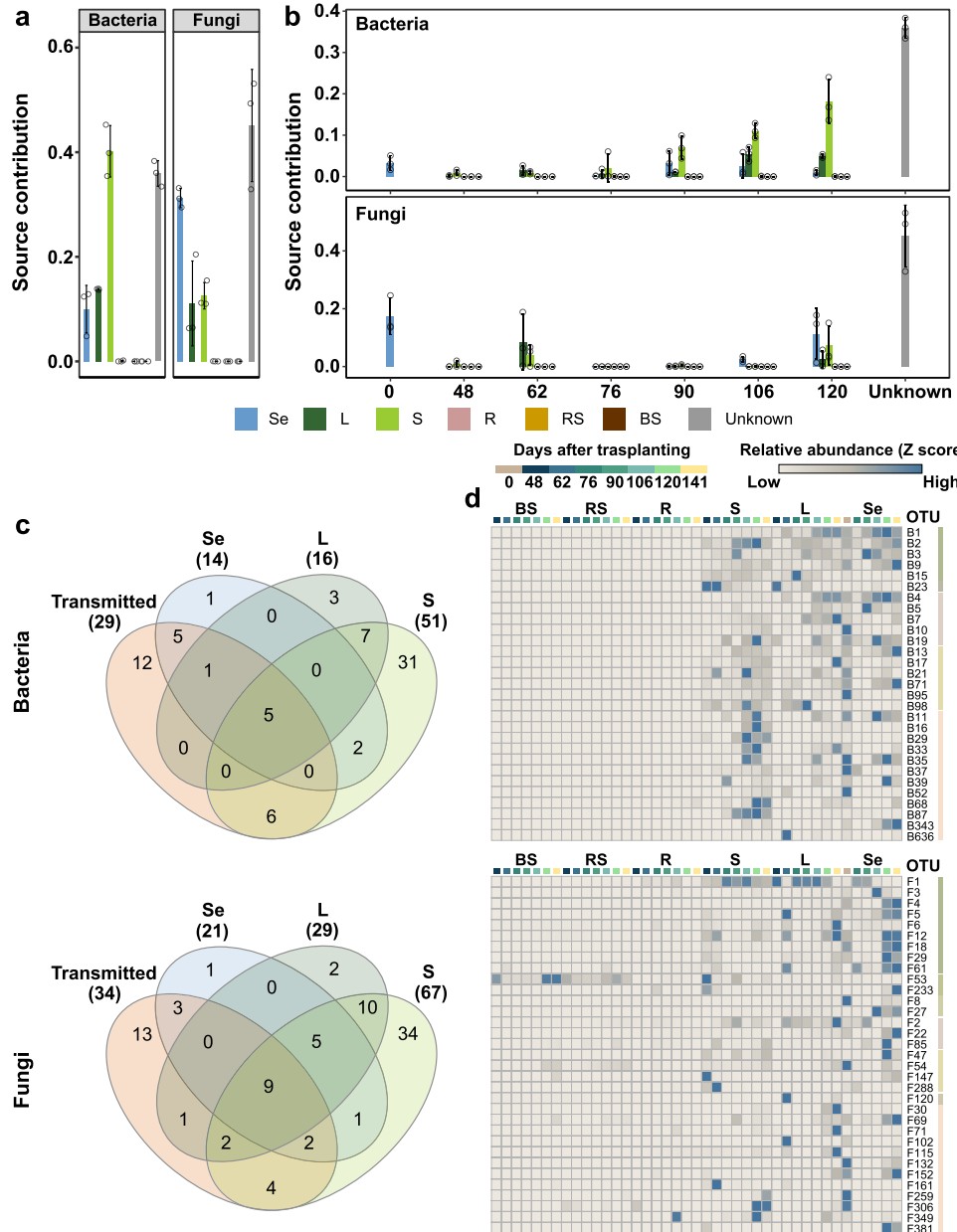

**Fig. 4 Source and possible route for the transmission of seed microbial community. a** Contribution of rice compartments to the formation of bacterial and fungal communities in progeny seeds at the harvest stage. **b** Contribution of rice compartments of each sampling time to the formation of bacterial and fungal communities in progeny seeds at the harvest stage. Error bars indicate the standard deviation of source contribution among plants. In **a** and **b**, dots indicate the estimated source contribution values in each repeat ($n = 3$). Colors of bars correspond to each rice compartment (Se [seed], light blue; L [leaf], green; S [stem], light green; R [root], light red; RS [rhizosphere], dark yellow; BS [bulk soil], brown; Unknown [Unknown source], gray). **c** Venn diagrams on vertically transmitted bacterial and fungal OTUs and core OTUs of bacterial and fungal communities in the aboveground compartments (leaf, stem, and seed). Core OTUs were identified as OTUs detected at 80% of prevalence in each compartment. The list of core OTUs is available in Supplementary Data 5. **d** Heat maps on the distribution of vertically transmitted bacterial and fungal OTUs across rice compartments. Heat maps were constructed with Morpheus software (https://software.broadinstitute.org/morpheus/). Relative abundances of each OTUs were z score-transformed. The bluer the color of the box, the higher the relative abundance. Colors of the boxes below the initials of rice compartments indicate sampling points (day after transplanting to field) when soil and plant compartment samples were collected. Colored bars next to the OTU labels correspond to the sections of the Venn diagrams in **c**. BS, bulk soil, RS, rhizosphere; R, root endosphere; S, stem endosphere; L, leaf endosphere; Se, seed; Transmitted, vertically transmitted OTUs.

homogenization of microbial associations throughout leaf, stem, and root.

**Microbial communities are not randomly associated during rice development**. Since we covered broad spectrum of microbial associations in the temporal and spatial scales, we evaluated whether

intra- and inter-kingdom associations are randomly formed or not. Using hypergeometric distribution analysis, a non-random distribution of intra- and inter-kingdom associations was examined (Fig. 5d, e). Intra- and inter-kingdom associations were mostly derived from the bacterial classes Alphaproteobacteria (orders Rhizobiales and Sphingomonadales) and Gammaproteobacteria

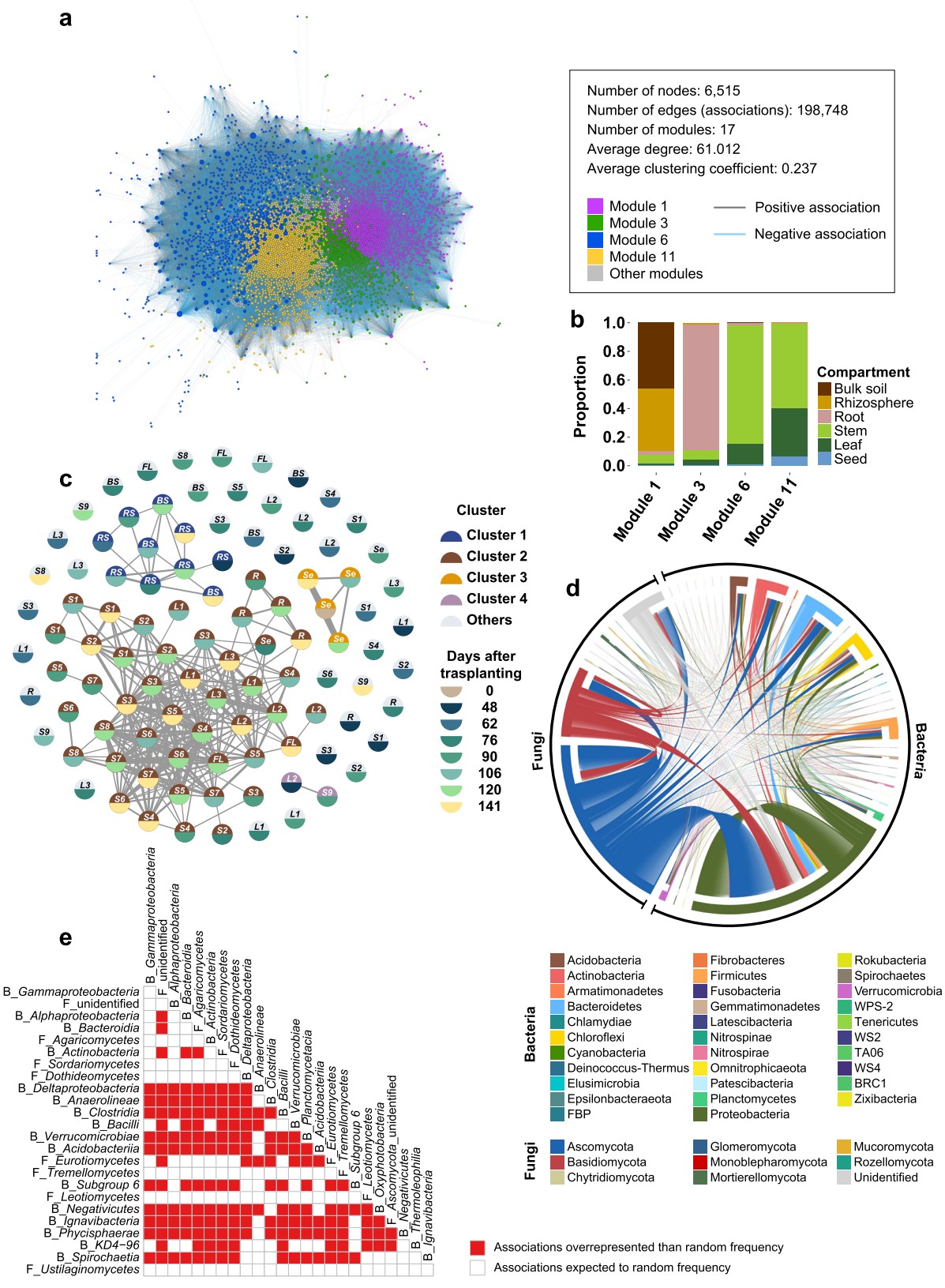

(orders Betaproteobacterales and Xanthomonadales) and fungal classes Dothideomycetes (order Pleosporales) and Sordariomycetes (orders Hypocreales and Sordariales) (Fig. 5d; Supplementary Data 9). Most of the associations between dominant bacterial and fungal classes and orders were overrepresented than random frequency (Fig. 5e; Supplementary Data 9). In the fungal classes

Sordariomycetes (order Hypocreales), Eurotiomycetes (order Eurotiales), Leotiomycetes (order Erysiphales), Agaricomycetes (order Polyporales), and Tremellomycetes (order Tremellales), bacterial-fungal associations showed non-random distribution but intra-kingdom associations did not. Microbial networks of each compartment showed the compartment-specific distribution

**Fig. 5 Global meta-network of rice-associated bacterial and fungal communities. a** Features of global meta-network of rice-associated bacterial and fungal communities across age and compartment. Node size is proportion to degree centrality of each node. Nodes are colored by modules (subnetworks) to which each node belongs. Pairwise associations are colored based on correlation coefficient values: positive (gray, correlation coefficient > 0) and negative (light blue, correlation coefficient < 0). **b** Proportion of associations consisting of top 4 modules in the global meta-network. The proportion of associations is calculated by dividing number of associations in each compartment by total number of associations in each module. Associations are colored by compartments where associations are detected. **c** Similarity of network structures based on Jaccard similarity index. Each node indicates bacterial-fungal co-occurrence networks in different compartments and ages. The colors of upper half circles of each node indicate clusters to which nodes belong, and those of bottom half circles of each node represent age of rice (days after transplanting). Links in which Jaccard similarity values are above 0.02 are displayed. The size of links is proportional to the value of Jaccard similarity index. BS, bulk soil; RS, rhizosphere soil; R, root endosphere; FL, flag leaves next to panicles; L3, the leaves at 30–40 cm from soil; L2, the leaves at 20–30 cm from soil; L1, the leaves at 10–20 cm from soil; S1–S9, stem samples which were separated at 10 cm intervals depending on the height of plants (S1 [stem at 0–10 cm from soil]-S9 [stem at above 80 cm from soil]); Se, seeds. **d** The taxonomic profiles of bacterial-fungal co-occurrence associations of dominant phyla and classes. Interconnections of bacterial and fungal taxa are indicated as edges in a directed network. Edges are lines that connect source node taxa with which edges start and target node taxa with which edges end. Edges are colored by the taxon acting as a source node. **e** Non-random distribution of bacterial-fungal associations across age and compartments at the class level. Overrepresentation of microbial associations was estimated using the hypergeometric distribution analysis. Kingdoms, where each class belongs, are indicated as "B_" for bacteria and "F_" for fungi. Overrepresented associations are colored by red squares, whereas randomly distributed associations are colored by white squares.

of associations (Supplementary Result 3; Supplementary Fig. 12; Supplementary Data 10). A greater number of dominant classes also showed associations represented more than random frequency in the belowground compartments than in the aboveground compartments. These results suggest that microbial associations in the aboveground compartments are more variable than in the belowground compartments on the temporal scale.

## Discussion

During plant growth and development, host and environmental factors affecting microbial communities are simultaneously changed. These changes can influence composition and functions of plant-associated microbial communities, further affecting the physiology of plants. Therefore, understanding on temporal dynamics of plant-associated microbial communities during plant development is crucial for harnessing microbiomes to improve crop productivity. Based on temporal and spatial tracking and comprehensive bioinformatic analyses, in this context, we revealed temporal-spatial dynamics of bacterial and fungal community structure and associations during rice development. Taking it a step further, we figured out the possibility of vertical transmission, potential sources, and transmission routes of seed bacterial and fungal communities to improve the understanding of the assembly of plant microbial community.

Using a plant model for tracking host-associated bacterial and fungal communities gives us advantages to identify vertically transmitted OTUs and their distribution in plant compartments during the life cycle of a host plant. We found 29 out of 15,975 bacterial (0.18% of total bacterial OTUs) and 34 out of 5963 fungal (0.57% of total fungal OTUs) OTUs potentially transmitted from generation to generation via seed (Fig. 2a; Supplementary Data 2). Community dissimilarity analysis showed that the abundance of vertically transmitted OTUs is restored during seed development (Fig. 2b). Linear regression analysis showed that most of vertically transmitted OTUs are late successors which abundances increased at the late developmental stage (Fig. 2c). These results suggest that temporal changes in seed environment may govern the colonization of vertically transmitted OTUs in developing seeds. During seed development, microbial communities encounter osmotic pressure and water stress due to the accumulation of starch and the loss of moisture (desiccation)[36,37]. Reactive oxygen species (ROS)-mediated programmed cell death occurring in 12 days after fertilization may also affect composition of seed microbial communities[36]. Thus, dramatic changes in ecological niches in seeds may act as a selection pressure against seed bacteria and fungi. Considering

that increase in the relative abundance of vertically transmitted OTUs as a late successor, it can be inferred that vertically transmitted OTUs may be able to endure environmental stresses or environmental changes may be favor for proliferation of vertically transmitted OTUs. Further investigation of environmental preference of seed-borne bacteria and fungi may show potential driver(s) involving the formation of seed microbial communities.

Conventionally, heritability analysis has been conducted to investigate the effects of genetic differences in a population on traits of a population[20], whereas the neutral model has been used to predict species-level neutrality in microbial communities at the spatial scale[22]. In the present study, we performed these analyses to investigate the effects of temporal changes in host physiology on the assembly of vertically transmitted OTUs in seeds. Heritability analysis demonstrated that vertical transmission of seed bacterial and fungal communities is partially under host control (Fig. 3a). Neutral model revealed that both neutral and niche processes affect the assembly of vertically transmitted seed bacterial and fungal communities during seed development (Fig. 3b). These results suggest that niche-based host selection and neutral effect contributed to the vertical transmission of seed bacterial and fungal communities. The effect of both selective and neutral forces on the vertical transmission of microbial community was also reported in wild sponges[38]. In the present study, the low heritability (low niche responsiveness) and contribution of neutral process may be driven by dormancy. Given that a mature seed is a dormant tissue and harsh environments for microbes, bacterial and fungal communities may be also in the inactive state that microbes make invisible to selection processes[39]. This highlights that dormancy may be important trait of vertically transmitted microbes for vertical transmission via seeds.

Combined analysis using heritability and Sloan's neutral model also discovered vertically transmitted OTUs that were governed by selective forces as which host control may act. In the bacterial community, OTUs belonging to *Pantoea*, *Sphingomonas*, *Methylobacterium*, and *Pseudomonas* were identified as the vertically transmitted OTUs under host control. A previous study reported that *Sphingomonas melonis*, a seed endophyte in rice, acts as a biocontrol agent against *Burkholderia plantarii*, a seed-borne pathogen[40]. *Methylobacterim* spp. have been reported to show growth-promoting activity in potato[41] and oil seed rape[42]. Meanwhile, in the fungal community, OTUs belonging to *Moesziomyces*, *Pyrenochaetopsis*, order Pleosporales, and family Phaeosphaeriaceae were identified. *Moesziomyces* spp. are closely related to smut fungi, *Ustilago maydis*. A recent study showed that *M. ballatus ex Albugo* antagonizes *Albugo laibachii*, the

oomycotal pathogen of *Arabidopsis*[43]. Given plants' preference that transmit beneficial endophytes via seeds[14], vertically transmitted OTUs identified in the present study may show beneficial effects (biocontrol activity or growth promotion) on rice. Further isolation and functional characterization of these bacterial and fungal OTUs will reveal the functional properties and effects of vertically transmitted OTUs under host control on rice physiology and development.

Bacterial and fungal communities of parent seeds and stem endosphere were primary source for microbial communities of progeny seeds (Fig. 4). Previous study showed that seed-borne bacterial and fungal communities can move from seed to seedling in the axenic conditions in rice[44]. Seed bacterial and fungal communities of common oak (*Quercus robur* L.) could be transmitted to phyllosphere and root of the developing seedlings under axenic conditions[45]. Presence pattern of vertically transmitted OTUs in stem sections demonstrated that vertically transmitted OTUs could move upward following the growth of stem (Supplementary Fig. 9). The unexpected result is that soil and root endosphere barely contribute to the formation of seed bacterial and fungal communities. In maize, root endosphere were important source for seed bacterial (39.6%) and fungal (28.4%) communities[26]. The contribution of soils to the formation of progeny seed bacterial and fungal communities was also reported in *Cucurbita pepo*[10]. These contrasting results suggest that the possibility of horizontal transmission of microbes from soil to seed may be low in rice. This difference may be driven by the existence of epiphytic microbes. In the study of *Cucurbita pepo*[10] and maize[26], surface sterilization was not conducted for preparing seed samples, indicating that both endophytic and epiphytic bacterial communities are included in the community data. The inclusion of epiphytic bacterial community may drive the different contributions of soils, including rhizosphere, to seed bacterial community. Another possibility is the existence of fruit that covers seeds. Rice seeds are covered with thick and dried dead tissues as coats, whereas seeds of *Cucurbita pepo* are exposed to moisture[46] that may support the growth and persistence of viable microbial communities. These environmental differences between rice and *Cucurbita pepo* may involve the differences in origin of seed microbial communities. An unknown source for progeny seeds suggests horizontal transmission of microbes from uncharacterized environment(s) to seed (Fig. 4a, b). In maize, 7.2% and 9.1% of seed-associated bacterial and fungal communities originated from air[26]. Similar to maize, microbes transmitted from air may consist of seed bacterial and fungal communities. Another uncharacterized source is paddy water. In the heading stage when the rice panicle is exposed to the external environment, paddy fields are submerged in 3–4 cm of water. This environmental characteristic of paddy field suggests that microbes in water may colonize in seed by splash or aerosols during seed development.

A meta-network and bacterial-fungal co-occurrence networks revealed dynamics of microbial associations over time and space. Previous studies have shown environmentally driven modules in microbial co-occurrence networks by different niches[47,48]. The modularity of a meta-network and similarity of microbial associations suggest differences in niche environments surrounding rice-associated microbial communities by plant compartment and age (Fig. 5a–c). The differences in the composition of modules of aboveground endophytic communities by host age suggest changes in niche environment in the endosphere of aboveground compartments between vegetative and reproductive stages. A previous study reported that root bacterial community stabilize compositionally in the reproductive stage[29]. The similarity of microbial associations of bacterial-fungal co-occurrence networks in the leaf, stem, and root endophytic communities after

reproductive stage suggests systemic stabilization of endophytic communities by similar niche environments across rice compartments. Aging-related host changes may involve the systemic stabilization of microbial communities in the aboveground compartments. During aging of rice, chloroplasts are dismantled, causing the production of ROS which lead to oxidative stress and damage to plant cells[49]. Further longitudinal analyses of transcriptome and metabolome will help identify host factors involved in stabilization of endophytic microbial communities.

Non-random distribution of microbial associations across rice compartments showed that microbial associations predicted in the aboveground compartments are more variable than in the belowground compartments in the temporal scale (Fig. 5d, e). Temporal variations in bacterial and fungal communities were also found in the ordination analyses (Fig. 1b, c). These findings suggest that microbial communities in the aboveground compartments may be more responsive to temporal changes in abiotic and biotic environments during rice development. It can be inferred that microbes colonizing in the phyllosphere may have delicate sensing systems for recognizing changes in host-driven niche environments. Further examination on genomic and transcriptomic characteristics on phyllosphere microbial communities will demonstrate key microbial factors leading microbes to adapt plant host system.

In conclusion, our work demonstrated compositional shift and ecological mechanism of vertical transmission of bacterial and fungal communities and dynamics of microbial associations in rice compartments, including endosphere of leaves, stems, and roots, bulk soil, rhizosphere, and seeds during rice development. In particular, understanding on temporal dynamics of metabolic status of both host plants and microbial communities will help identify key drivers affecting the assembly of microbiomes and unveil key microbial functions related to plant fitness. In this context, our study provides ecological cornerstones on host-microbiome relationships affecting plant fitness during plant development.

## Methods

**Collection of soil and plant samples in 2017**. To track the compositional variations of rice-associated microbial communities according to the growing season and geography, three fields located in two geographically distant locations were selected (CC1, 37°54′14.9″N 127°44′02.1″E; CC2, 37°54′16.4″N 127°44′01.3″E; SW, 37°16′07.1″N 126°59′23.5″E). The fields were differently managed. A total 9 kg of nitrogen, 4.5 kg of phosphate, and 5.7 kg of potassium per 1000 m² were applied three times per cultivation period in the two fields (SW and CC1). The other field (CC2) was also fertilized in the same way but any pesticides were not applied during rice cultivation. In 2017, three cultivars were used for further works. Gohyangchal (faster-growing rice), Daean (slower-growing rice), and Akibare (slower-growing rice) that are generally cultivated in each geographic site were cultivated in CC1, CC2, and SW, respectively. Plant and soil samples were collected from three plots of each field during rice cultivation. Rice that did not show any visible symptoms of diseases were obtained to exclude the effects of pathogens on endophytic bacterial and fungal communities. For collecting bulk soils, to prevent the effects of plant residues and roots, topsoils (0–5 cm depth) between rice plants were removed using a sterilized shovel. One kg of soils below a depth of 15 cm was collected. A total of 36 bulk soil samples (3 fields × 3 plants × 4 sampling times) were collected. Among the collected soils, 500 g of each soil was sieved with a 2 mm mesh to remove remaining plant debris and particles that are bigger than sand. 0.5 g of sieved soils were transferred to Lysing Matrix E tubes provided in FastDNA SPIN Kit for Soil (MP Biomedicals, Ohio, USA). The transferred samples were kept at −80 °C until DNAs were extracted. A total of 27 plant samples (3 fields × 3 plants × 3 sampling times) were collected from the same plots where the soil samples were acquired. Among plant compartments, seed samples except other plant compartments were collected at the harvest stage. Whole rice plants were obtained and were carried in the ice to prevent any changes in microbial communities. The collected rice plants were dissected into each compartment: leaves, stems, roots, and seeds. Since there was a difference in the growth rate among cultivars, different numbers of seed samples were acquired according to cultivars in the same sampling period (Sampling point when seeds were collected: CC1, 80 and 120 days after transplanting; CC2, 120 and 140 days after transplanting; SW, 80, 120, and 140 days after transplanting). A total of 27 seeds (9 seeds per rice plant) were obtained in each sampling point. Because a

single seed showed lower DNA concentration compared to other compartment samples, three seeds were pooled into a single Lysing matrix S tube (MP Biomedicals, Ohio, USA) for further DNA extraction. Therefore, a total of 144 rice compartment samples (leaves, $n = 27$; stems, $n = 27$; roots, $n = 27$; seeds, $n = 63$ [CC1, $n = 18$; CC2, $n = 18$; SW, $n = 27$]) were obtained. Roots below a depth of 15 cm were acquired. The divided compartments were surface-sterilized to eliminate the epiphytic portion of microbial communities. Surface sterilization was conducted by sequential treatment of 70% EtOH and 2% NaClO (70% EtOH for 2 min, 2% NaOCl for 2 min, and 70% EtOH for 1 min). After the treatment, tissue samples were washed using sterilized distilled water at least ten times for removing residual chemicals. After the surface sterilization, remaining water of the washed samples was removed using sterilized miracloth. Then, the treated compartments were ground using sterilized mortar and pestles. 0.5 g of the ground plant samples were further transferred to Lysing Matrix E tubes. The transferred samples were kept at −80 °C until DNAs were extracted. Among the remaining seeds that were harvested in the SW field were dried at 37 °C for 1 week and stored at 4 °C to use them for growing 2018 samples.

**Collection of soil and plant samples in 2018.** To identify the temporal changes in microbial compositions in detail, we collected rice plants from the field SW in the same manner with the marginal modification. Seeds harvested and stored in 2017 were used for sowing. The rice plants in the field SW (cultivar Akibare) were cultivated in the same manner in 2017. The sampling started 48 days after transplanting into the field. The rice and soil samples were collected every two weeks (seven sampling points). The sampling field was divided into three plots, and one plant was collected in each plot (1 field × 3 plants as biological replicates × 7 sampling times). Bulk soils ($n = 63$ [21 biological replicates × 3 technical replicates]) were collected in the same manner used in the 2017 samples. For collecting rhizosphere samples ($n = 63$ [21 biological replicates × 3 technical replicates]), we adopted the collection scheme from the work of Edwards et al.[50]. Roots with soils were collected from rice plants, loosely attached soils were removed to leave 1 mm of soils around the roots. The roots with 1 mm of soil were transferred to 50 ml tubes. The soils attached to roots were washed off in phosphate-buffered saline solution (pH = 7.4) using a vortex mixer at the highest speed. After transferring the washed roots to clean 50 ml tubes, the mixtures were centrifuged for 5 min at 10,000 $g$ to concentrate soils. The supernatant was discarded, and the remaining soils were used as rhizosphere samples. 0.5 g of the collected rhizosphere samples were transferred to Lysing Matrix E tubes. Of the remaining roots (n = 63 [21 biological replicates × 3 technical replicates]), 5–15 cm below the ground were used for DNA extraction. Technical replicates for soil and root samples were obtained by transferring 0.5 g of soil or ground root samples to three individual tubes for DNA extraction. A total of 21 plant samples (1 field × 3 plants × 7 sampling times) were collected from the same sampling sites where soil samples were obtained. As described in the belowground compartments, leaf and stem samples also included three technical replicates per biological replicate. Leaf samples were divided into subgroups based on the position of leaves (FL, flag leaves which are located next to panicles [$n = 45$]; L3, the leaves located at 30–40 cm from soil [$n = 54$]; L2, the leaves located at 20–30 cm from soil [$n = 63$]; L1, the leaves located at 10–20 cm from soil [$n = 63$]). Stem samples were also separated at 10 cm intervals depending on the height of plants (S1, stem endosphere located at 0–10 cm from soil [$n = 63$]; S2, stem endosphere located at 10–20 cm from soil [$n = 63$]; S3, stem endosphere located at 20–30 cm from soil [$n = 63$]; S4, stem endosphere located at 30–40 cm from soil [$n = 54$]; S5, stem endosphere located at 40–50 cm from soil [$n = 45$]; S6, stem endosphere located at 50–60 cm from soil [$n = 42$]; S7, stem endosphere located at 60–70 cm from soil [$n = 36$]; S8, stem endosphere located at 70–80 cm from soil [$n = 30$]; S9, stem endosphere located above 80 cm from soil [$n = 21$]). The separated compartments were further surface-sterilized for removing epiphytic microbial communities in the same manner for treating the samples collected in 2017. Then, the sterilized plant compartments were ground using sterilized mortar and pestles. 0.5 g of the ground samples were transferred to Lysing Matrix E tubes. In this step, three technical replicates were obtained for leaf and stem samples. A total of 459 seed samples as biological replicates were collected at the sowing stage and from heading to harvest stages (sowing stage, $n = 81$; booting stage, $n = 54$, heading stage, $n = 81$; dough stage, $n = 81$, yellow-ripening, $n = 81$, harvest, $n = 81$). Seed samples were treated in the same way used for 2017 samples. All samples (153 samples) were kept at −80 °C until DNAs were extracted. Meta data and information on developmental stages corresponding to each sampling point are available in Supplementary Data 11.

**Sample preparation and DNA extraction.** DNAs of all samples were extracted using FastDNA SPIN Kit for Soil (MP Biomedicals, Ohio, USA). The collected and processed soil and rice compartment samples were prepared and pulverized using a bead beater (Biospec Products, Oklahoma, USA) at 4000 rpm for 2 min. This step was repeated after cooling in ice for 1 min. Soil and plant DNAs were extracted following the instructions of the manufacturer. The concentration of DNA samples was quantified by NanoDrop™ spectrophotometers (Thermo Fisher Scientific, Massachusetts, USA). The extracted DNAs were stored at −20 °C until amplicons were generated.

**PCR amplification and sequencing.** 16S rRNA and ITS amplicons were generated in a two-step PCR amplification protocol. The V4 regions of bacterial 16S rRNA genes were amplified with universal 515F and 806R PCR primers (Supplementary Table 1)[51]. To reduce plant mitochondrial and plastid DNA contamination, peptide nucleic acid (PNA) PCR blockers (Panagene, Daejeon, Republic of Korea) were added during the first PCR (Supplementary Table 2)[52]. The fungal ITS2 regions of the 18S ribosomal RNA genes were amplified by ITS3 and ITS4 PCR primers (Supplementary Table 1)[53]. Each sample was amplified in triplicate in a 25 μl reaction tube containing 12.5 μl of 2 × PCR i-StarTaq™ Master mix solution (iNtRON Biotechnology, Seongnam, Republic of Korea), 0.4 μM for each forward and reverse primers, 0.8 μM of diluted DNA template and PNA clamps for chloroplast and mitochondria at 0.75 μM each. For the ITS libraries, the conditions were the same except the PNA clamps were not included. PCR was performed using the following program, initial denaturing at 98 °C for 3 min, followed by 32 cycles of denaturing at 98 °C for 10 s, PNA annealing at 78 °C for 10 s, primmer annealing at 55 °C for 30 s and extension at 72 °C for 60 s. For ITS PCR amplification, the program was the same but without the PNA annealing step. Each library was accompanied by negative PCR controls to ensure that the reagents were free of contaminant DNA. Amplicon replicates were pooled, then purified using the MEGAquick-spin™ Plus DNA Purification Kit (iNtRON Biotechnology, Seongnam, Republic of Korea) with an additional ethanol clean-up step to remove unused PCR reagents and resulting primer dimers. Secondly, the PCR was done with the Nextera XT Index Kit (Illumina, California, USA). DNA templates were diluted to equal concentrations after being measured by the Infinite 200 pro (Tecan, Männedorf, Switzerland). The libraries were then pooled into equal concentrations into a single library and concentrated using AMPure beads (Beckman Coulter, California, USA). The pooled library then went through a final gel purification stage to remove any remaining unwanted PCR products. Pooled libraries were sequenced using the Illumina MiSeq platform with 2 × base pair read length. The sequencing was done in the National Instrumentation Center for Environmental Management at Seoul National University, Republic of Korea.

**Processing of microbial reads, and statistical analyses on microbial communities.** The sequenced reads were further processed with QIIME2 (version 2020.2)[54]. After demultiplexing, the resulting sequences were merged and then quality filtered using DADA2 plugin[55] in the QIIME2 (version 2020.2) pipeline. The high-quality sequences were clustered into OTUs using the open reference vsearch algorithm (vsearch cluster-features-open-reference)[56] against the Silva 99% OTU representative sequence database (v132, April 2018)[57] and then assembled into an OTU table. Bacterial OTUs were chimera filtered using the vsearch uchime-denovo algorithm[58]. Fungal OTUs were checked for chimeric sequences using Uchime-ref algorithm against the dedicated chimera detection ITS2 database (June 2017 version)[59]. After the OTU clustering, the taxonomy of the non-chimeric OTUs was assigned using Naïve Bayes algorithm implemented in the q2-feature-classifier prefitted to the Silva database for V4 region of 16S rRNA regions[60]. For the ITS2 region, taxonomy assignment was done with q2-feature-classifier prefitted to UNITE database (UNITE_ver7_dynamic of Jan 2017)[61]. Bacterial sequences with a length over 300 bp and fungal sequences less than 100 bp long were discarded. The OTU table was imported to R by the phyloseq package[62] for further analysis. Sequences from host DNA and OTUs unassigned at the kingdom-level were removed (bacterial OTU: orders Chloroplast and Rickettsiales that were derived from rice DNA; fungal OTU: kingdoms Unassigned, Chromista, and Plantae). OTUs detected from negative controls were removed from the samples.

**Statistical analysis and visualization.** Unless otherwise stated, all statistical analyses were performed using R version 3.5.2[63] and statistical significance was determined at $a = 0.05$, where appropriate, the statistical significance was corrected for multiple hypothesis testing using the false discovery rate (FDR) method. The OTU table was normalized by cumulative-sum scaling (CSS) and log-transformed by the function *cumNorm* from the R package metagenomeSeq (v 3.8)[64]. Normality was tested using Shapiro-Wilk normality test. Kruskal-Wallis test and two-sided Dunn's test were all performed in R. Taxa above relative abundance of 0.5% were visualized with the R package ggplot2 (v3.2.1)[65] for taxonomic composition analysis. The Bray–Curtis dissimilarity matrix was calculated to build principal coordinate analyses. PERMANOVA was conducted using the function *adonis* from the R package vegan (v2.5.5)[66]. In the PERMANOVA analysis, factors including plant compartments, rice age (sampling timepoint), developmental stage, soil conditions, air temperature, geographic locations were considered. Community dissimilarity between seeds at the sowing stage and developing progeny seeds was calculated using Jaccard distance. Vertically transmitted seed OTUs were defined as OTUs detected in all rice seeds of harvest stage in 2017 and seeds of pre-cultivation and harvest stages in 2018. Remaining OTUs of seed bacterial and fungal communities were considered as transient OTUs that could colonize in seeds but not persist across generations. Core OTUs of rice compartments were identified at the 90% of prevalence across all soil sites using the function *prevalence* implemented in the R package microbiome (v1.9.13)[67].

**Heritability analysis and Sloan's neutral model on seed bacterial and fungal communities.** To find the effect of temporal changes in niche environments on

temporal changes in OTU abundances in seeds, we adopted heritability analysis. We considered niche environments according to rice ages as an analog of genetic influences that affect plant phenotypes. For this analysis, OTUs which relative abundance is lower than 0.0001 were removed from OTU abundance table. Heritability of each OTU was estimated by a linear mixed-effects model with 100 bootstraps using the function *lmer* implemented in the R package lme4[68]. Rice age and biological repeat were considered as a fixed effect and random effect on centered log-ratio-transformed abundances of OTUs, respectively. OTUs were grouped based on the heritability estimates ($h^2$). OTUs which $h^2$ is over 0.4 were grouped as "High", whereas those which $h^2$ is less than 0.2 were classified as "Low". Others were grouped as "Moderate". Since heritability concept could not show the effect of neutrality on the temporal assembly of seed bacterial and fungal communities, we assessed our data for the suitability of the Sloan's neutral model[22]. This model predicts relationships between the occurrence frequency of taxa in local communities and their abundance across a broad metacommunity. In this analysis, we considered seed bacterial and fungal communities identified at each time as local communities, except seed at the sowing stage ($n = 126$). Since this study spanned 140 days, we considered that microbial speciation and diversification processes are unlikely to have a meaningful contribution to assembly. Thus, their effects were excluded from the model. We used the R script made available by Hassani et al.[69]. The Sloan's neutral model was fit for observing the occurrence frequency of OTUs in local communities and their abundance in a metacommunity with the migration rate ($m$), which was interpreted as the influence of temporal dispersal.

**Identification of sources of seed microbial communities**. To find the source of seed microbial communities, we examined sink–source relationships between seed samples and other plant and soil compartments. For this analysis, we used a fast expectation-maximization for microbial source tracking (FEAST) algorithm[70]. Bacterial and fungal communities in seeds collected at the harvest stage (141 days after transplanting) in 2018 were considered as sink samples. Microbial communities of leaf, stem, root, rhizosphere, and bulk soil collected at the other stages were denoted as sources. Two hyperparameters, convergence threshold and the maximum number of iterations, in FEAST were set to $1 \times 10^{-6}$ and 1, respectively.

**Construction and analysis of bacterial-fungal co-occurrence networks**. Multi-kingdom co-occurrence networks were constructed to infer the variations in microbial associations of rice compartments during the life cycle of rice. In order to construct the co-occurrence networks of each developmental stage, CSS-normalized multi-kingdom OTU tables were used as an input for the network construction. The SparCC algorithm[71] was used to infer co-occurrence patterns[71]. Significant correlations were defined with correlations which coefficients >0.3 or <−0.3 with FDR-adjusted $P < 0.05$. Visualization was performed with Gephi (v0.9.2)[72] using the Fruchterman Reingold layout. The node, edge, and network properties (degree, betweenness centrality, closeness centrality, and clustering coefficients) were investigated with igraph[73]. Hub OTUs were simply defined as 10 nodes with the highest degree to correct for biases of sample or taxa number[26]. Hub node profiles were clustered by hierarchical clustering based on Jaccard distance implemented in Morpheus (https://software.broadinstitute.org/morpheus/). To quantitatively compare network complexity among seed microbial networks, it was measured via three different indices including Bertz complexity index, Atom-bond connectivity, and Geometric-arithmetic index using the R package brainGraph (version 2.7.3)[74].

**Construction and analysis of a global meta-network**. To construct a global meta-network, 101 bacterial-fungal co-occurrence networks were merged. Visualization of the global meta-network was performed with Gephi using the Force Atlas 2 layout. The node, edge, and topological properties were investigated in the same way as bacterial-fungal co-occurrence networks of each compartment and age. Modularity analysis on the global meta-network was performed using the algorithm implemented in Gephi[72]. To find overrepresented associations (non-randomly distributed associations) across compartment and age, hypergeometric distribution analysis was performed using the function *phyper* implemented in R with the R script made available by Ma et al.[32]. Taxon–taxon co-occurrence and co-exclusion analysis across dominant taxa were conducted using the binomial distribution using the R function *pbinom*. In all analyses, $P$ values were adjusted with FDR using the R function *p.adjust*.

**Similarity of bacterial-fungal co-occurrence networks across compartments and age**. The similarity of network structures across compartments and age was examined using Jaccard similarity index. Jaccard similarity values were calculated by dividing the count of overlapped associations between two networks by total counts of associations in the same network pair. The Jaccard similarity matrix was visualized by a network using the function *graph_from_adjacency_matrix* implemented in the R package igraph. The network was finally plotted with Gephi using the Fruchterman Reingold layout.

**Statistics and reproducibility**. As stated in the Methods, all statistical analyses were performed using R version 3.5.2 and statistical significance was determined at

$P < 0.05$, where appropriate, the statistical significance was corrected for multiple hypothesis testing using FDR method. A total of 48 rice plants (27 plants for 2017 and 21 plants for 2018) and corresponding soils were collected as biological replicates. The collected plants were further divided into detailed compartments, including seeds, leaf, stem, root, and rhizosphere. Raw sequences were processed using QIIME2 (version 2020.2). Statistical analyses were performed using the R packages vegan (version 2.5.5), metagenomeSeq (version 3.8), microbiome (version 1.9.13) in the R software (version 3.5.2). All statistical analyses were performed based on biological replicates. More details on biological and technical replicates and statistical analyses are available in the "Methods" and figure legends.

**Reporting summary**. Further information on research design is available in the Nature Research Reporting Summary linked to this article.

## Data availability

Source data underlying figures in the main text are available in Supplementary Data 1, 2, 3, 4, 5, 6. Source data underlying supplementary figures are available on GitHub (https://github.com/hyunkim90/spatiotemporal_tracking_rice_endophytic_communities). All raw sequences derived from this experiment were submitted to the Sequence Read Archive of NCBI and can be found under the BioProject accession numbers PRJNA728671, PRJNA728647, and PRJNA733292.

## Code availability

Metadata files, R data files, and R notebooks for full analyses are available from https://github.com/hyunkim90/spatiotemporal_tracking_rice_endophytic_communities.

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

## Acknowledgements

This work was supported by the National Research Foundation of Korea (NRF) grants funded by the Korean government (MSIT) (2020R1A2B5B03096402, 2018R1A5A1023599, and 2021M3H9A1096935 to Y.-H.L. and 2022R1C1C2002739 to H.K.) and Korea Institute of Planning and Evaluation for Technology in Food, Agri-culture, and Forestry through Agricultural Microbiome R&D Program, funded by Ministry of Agriculture, Food, and Rural Affairs (MAFRA) (918017-04).

## Author contributions

H.K. and Y.-H.L. conceived and designed the study. H.K., K.K.L., and J.J. collected the samples. H.K. performed all analyses. H.K. and Y.-H.L. analyzed the data. H.K.

contributed to the writing of the manuscript. H.K. and Y.-H.L. prepared the manuscript. All authors read and approved the final manuscript.

## Competing interests

The authors declare no competing interests.
