## [Peer Review File · Communications Biology]

Reviewers' comments:

Reviewer #1 (Remarks to the Author):

The article has substantially improved and authors have politely and thoroughly answered all my and other reviewers' comments. I only have a minor statistical question on handling the technical replicates which became apparent only after revisions: were the biological and technical replicates treated equally in the statistical models (i.e. PERMANOVA and networks?). Usually one would only include the information of biological replicates in the models as technical replicates are pseudoreplicates and obscure the patterns. The authors should check this and run the relevant models with only using biological replicates to check that the pattern detected is robust enough and biologically relevant. As variation is not shown in most figures, it should not affect the figures too much. If the technical replicates are already omitted or averaged, this is ok and you can ignore this comment.

I have few tiny corrections further

Line 62: maybe add 'open' before or 'remaining' after 'questions'

Line 66: 'knowledge'

Hope my comments were useful, this is really a nice piece of work that is at the same time presented in a very clear and beautiful manner.

Kind regards,
Emilia Hannula

Reviewer #2 (Remarks to the Author):

The authors satisfactorily addressed my issues.

Point-by-point response to reviewers' comments

We very appreciate the insightful comments and suggestions on our manuscript. Our point-by-point responses to the comments can be found below.

All the changes made in the revised manuscript can be found in the uploaded original MS Word file.

Reviewers' comments:

Reviewer #1 (Remarks to the Author):

The study in question is performed to the highest standard. The study addresses a novel question on transfer of microbiomes in seeds. Especially the fungal part is novel. The sheer number of samples and the eye for detail is excellent. The data is visualized beautifully and clearly, which is not given fact when there is so much of it. The data provided corresponds to the data presented in the figures. There are some concerns on the set-up of the experiments (the caveats and low number of replicates and potential pseudo-reps should be acknowledged) but despite the differences in design the pattern observed is remarkably solid. The manuscript text is in itself too long and the main message of the paper could be explained in less words. I have indicated below which parts could be shortened/omitted.

1. The abstract should be improved. Now it in very short sentences states what was done and found out but lacks excitement. Based on this abstract I would not read the article. For example sentence 'late successors which abundances gradually increased from developing to developed seeds' is very important, yet without proper context it is very difficult to understand what it means. Further (line 30-31) it is unclear how you conclude that the succession is under host control (more details here needed).

'Bacterial succession was more affected by neutrality than fungi' – I assume the authors mean that fungi were not affected by neutrality while now it can be interpreted that bacteria are not affected by fungi. It should be further clarified what is meant with 'seed-to-seed transfer'

- Answer: As you commented, we revised the abstract to clarify the findings in this study with proper contexts that support the findings (Lines 31-43).

2. Introduction: the first few sentences should be references as well.

- Answer: References were added as you commented (Lines 46-49).

3. Lines 59-60 very long brackets, important point - consider revising. Nice that questions are clearly presented early on in the ms.

- Answer: We divided the too long sentence that you mentioned into three sentences (Lines 61-68).

4. Line 60 'sources' refers to what?

- Answer: The word "sources" refers to microbial pools where seed microbes originate (possible origin of seed microbial communities), such as soils and plant endosphere. We added the explanation on sources in the Introduction in the lines 66-68.

5. Line 63: niche and neutral processes.

- Answer: The words were corrected as you commented (Line 70).

6. Line 78: add 'that'.

- Answer: The sentence was revised (Line 85).

7. Line 81: add 'the'.

- Answer: The word "the" was added in the line 87.

8. Line 85-86: quite a basic sentence that is not needed here.

- Answer: The sentences were deleted as you commented (Lines 92-93).

9. Line 98: Maybe good to explain why 2 years selected (i.e. which question it answers).

- Answer: We investigated rice-associated bacterial and fungal communities across two consecutive years to identify bacteria and fungi that can be constitutively transmitted from seeds to seeds across years. This point was added in the Result section in the lines 117-120.

10. Line 100: again 'source' is unclear.

- Answer: As addressed above, "source" refers to microbial pools where seed microbes originate, such as soils and plant endosphere.

11. Line 102: something is missing in the sentence (or the last part is too much and unrelated?).

- Answer: We deleted the sentence that you mentioned since the sentence hampered the context of the paragraph (Lines 110-111).

12. Results: Line 108: Only one study cited so not 'studies'. In this sentence active tone would be better, microbes are vertically transferred in seeds. It would be good to explain better why during 2 years, why these plant parts and why there are 3 locations. And add how many replicates were analysed. Also, the number of total reads is not as interesting as reads per plant part.

- Answer: The sentence was revised as you commented (Line 116). Brief explanations on sampling strategy and replicates were added in the lines 117-128. We also added the information on reads per a plant part instead of total reads (Lines 136-139).

13. In the next section of results (starting from second paragraph) there are way too many details and numbers. These numbers should be visible in the figures and here only the statistical tests. Also, the acronyms are not explained so there is no way to interpret these results. It also seems main results are found in supplements as they are cited first. I would assume community structure would be the first thing (along with diversity) to report, here the authors go straight to individual taxa (phyla). The abundances are likely to be relative abundances, please be precise with wording. For fungi, phylum level is too coarse. Saying that there are Ascomycota somewhere does not tell anyone anything (it is the most common phylum in most terrestrial habitats making up 70% of fungal species). Similarly, Basidiomycota is the second most common phylum so finding them is not novel. I would suggest the authors dig a bit deeper into this data and report at least fungi (and why not bacteria) on order/class level. This is shown in figures (for fungi) but is not thus in line with text. Also, as the orders are mixed, one cannot see from the figure anything on Ascomycota vs. Basidiomycota (Tremellomycetes seem to be dominant Basidiomycota while dominant Ascomycota are Sordariomycetes).

- Answer: To make the manuscript more concise, we rewrote the paragraphs you commented (Lines 143-165). We also added the explanations on acronyms (Lines 163-

165). Too many details on relative abundance of bacterial and fungal taxa were removed. Description of relative abundances of fungal phyla was also removed. Instead, we added descriptions on taxonomic compositions at the class and order levels with supporting data (Supplementary Data 1) (Lines 143-165). Bacterial and fungal taxa were ordered by their relative abundances in each compartment.

14. Line 144-147: This suggests... is this not tested with stats here? This first part is very poor in statistical analysis. Beta diversity = community structure is better. Would be great to know also here what factors are considered in the (Permanova) models.

- Answer: We removed the sentence. Instead, we added the sentence "The observed compositional variations by compartment and age were supported by ordination analyses and permutational analysis of variance (PERMANOVA)." in the lines 195-198. Factors that were considered in PERMANOVA were added in the Method section (Lines 859-862).

15. Lines 157-162: Is this in time? There is no real need to show two separate distance indices showing the same but forget to explain what is compared.

- Answer: We used Jaccard and Bray-Curtis community dissimilarity indices to show the similarity in presence/absence patterns and abundance patterns of bacterial and fungal OTUs, respectively. We moved this part to the second section of Results entitled "Vertically transmitted OTUs dominate seed microbial communities during seed maturation" (Lines 239-247) since this part was not compatible to the first section of Results. Shortened descriptions for the analysis were added in the lines 239-247.

16. In the Figure 1 it would be good to know how many replicates there are – and what testing on phylum/class level abundances has been done.

- Answer: Replicate information was added in the caption of the Fig. 1 (Lines 219-224). We also described how relative abundances in Fig. 1a were estimated at each taxonomic level (Lines 219-224).

17. Is it the same experiment for the vertically transmitted OTUs. Are they included in the previous analysis already? This is a bit unclear.

- Answer: In the previous analysis, community dissimilarity was estimated using entire community members detected in parent seeds and progeny seeds. On the other hand, in this analysis, we only considered vertically transmitted OTUs (29 bacterial and 34 fungal OTUs) to investigate the relative abundance patterns and succession modes. To reduce the confusion, we revised the sentence by adding the phrase “which consisting of 29 bacterial and 34 fungal OTUs” in the lines 270-271.

18. L189-190: Transmitted OTUs belonging to bacteria... The part on transmitted OTUs is very nice. However, I wonder here at what level the transmission was confirmed. Was it per plant or averaged over plants?

- Answer: The transmission was confirmed from the average over plant replicates. The sentence was revised to clarify what you mentioned (Line 251).

19. The information on identity of the OTUs would be nice to have also in the figure 3 (instead of only mentioning it in the text).

- Answer: The information on the identify of OTUs was indicated in the revised Fig. 3 (Line 344) or attached below. The explanations on the OTU labels were added in the lines 356-358.

20. The part on sources of microbes is clear – but at this point reader starts to get tired as the results section is pretty long so it could be shortened considerably. Figure is very clear though.

- Answer: We shortened the result part regarding source-sink analysis as you commented (Lines 362-388).

21. The part on ‘Vertically transmitted OTUs compose core microbial communities of aboveground compartments’ is not probably needed at all but can be with few sentences incorporated into the previous sections. This is very descriptive and tiring to read. More information takes out from giving the main message of the study so I would really consider shortening.

- Answer: The part on ‘Vertically transmitted OTUs compose core microbial communities of aboveground compartments’ (Lines 425-449) was moved to Supplementary Result 1. Brief description of this part was added to the part on “Microbial communities of stem and parental seeds are major sources of those of progeny seeds” (Lines 390-397).

22. Although networks are well executed, I also doubt if they are needed as they are not directly answering the questions asked here. The message shown is pretty clear and this long section on networks is a nice visual but does not really offer too much to the manuscript. Of course, you can look at niche co-occurrence in compartments but that should be then clearly asked and be part of the story. Here as a note, fungi are analysed at different level than in figure 1 (now phyla).

- Answer: In the network part, we aimed to understand temporal and spatial dynamics of microbial associations during seed-to-seed transmission since microbes interact with each other in environments (Lines 452-453). Too many descriptions on network

similarity (Lines 483-495) were moved to Supplementary Result 2 in the revised Supplementary Information. The Fig. 5d was visualized at the phylum level because the figure would be complicated when it was visualized at the class or order levels. Instead, description at the class and order level was added in the lines 534-545. Supporting data at the class and order levels was also added in the Supplementary Data 9 and 10.

23. The part on non-random associations is actually probably going to show the same as the analysis on which groups were enriched in which compartment of the plant (first bit of the results). This is interesting but could be simplified (and again, more detailed level of analysis for microbes).

- Answer: Description at the more detailed taxonomic level (class and order) was added as you mentioned (Lines 534-545). Descriptions regarding non-random distribution of microbial associations in rice compartments (Lines 547-552) were moved to Supplementary Result 3 in the revised Supplementary Information.

24. Beginning of discussion resembles introduction.

- Answer: In the first paragraph of the discussion, we intended to summarize the present study. Because of this, we think that you felt the beginning of Discussion is similar to Introduction. To reduce the redundancy, we rephrased the paragraph (Lines 559-576).

25. Line 476: what is 'whole plant' here?

- Answer: "Whole plant compartments" means all the plant compartments consisting of the plant body. The sentence was removed since we rephrased the paragraph to

which the sentence belonged (Lines 559-576).

26. Line 484-485: maybe mention that they made up large portion (and not only the number...). Would be interesting to know the % they make up from total OTUs as well. Is 29 a lot from 10 000?

- Answer: 4.2% and 5.39% of bacteria (686 OTUs in developing and developed seeds) and fungi (630 OTUs in developing and developed seeds) were identified as vertically transmitted OTUs. When considered all OTUs in this study, 0.18% for bacteria (29 out of 15,975 OTUs) and 0.57% for fungi (34 out of 5,963 OTUs) could be vertically transmitted. Therefore, we think that the limited portion of microbes can be transmitted from seeds to seeds. The portion of vertically transmitted OTUs was added in the lines 579-580.

27. Methods: The use of different cultivars AND locations should be explained earlier as well (they are referred to). Is it that in each site different cultivar was used? It is amazing how little the site management affects the microbes actually which is nice but the design is not really optimal in this way. Were the factors site and cultivar also added to the analyses?

- Answer: In each site, different cultivars were grown. The cultivars and different locations were included in community diversity analyses (taxonomic composition and compositional variance). For analyses on vertically transmitted microbes, rice plants that were collected from one field where cultivar Akibare was grown in 2017 and 2018 to exclude the effects of geographic sites and cultivars.

28. 3 is pretty little replicates especially when no pooling between plants were used. This is then really individual plants used. The forward selection of not including sick plants should be also noted earlier as they would have very different microbiomes. In the design 3 fields x 3 plants is it really plants or plots. How was the soil collected?

- Answer: In this study, we collected three individual plants per a field. Individual plants consisted of tillers that were differentiated during the vegetative stage of rice. Enough amounts of plant samples were acquired from the tillers. The forward selection of not including sick plants was noted in the line . Using a sterilized shovel, topsoils (0-5 cm depth) were removed and 5-15 cm of soils that were not affected by rice roots were collected (Lines 717-718).

29. In the second experiment how were the technical replicates handled in the analysis? The amount of 3 biological replicates is really little – what was done if sample failed?

- Answer: As described above, plant samples were acquired from tillers composing individual plants. Leaf and stem samples located in the same positions in tillers were pooled for extracting DNAs. To extract DNAs from soils and plant compartment samples, 0.5 g of samples were used. For plant compartment samples, each sample was ground first using sterilized mortars and pestles. We could acquire 3-4 g of ground samples in this step. As a result, we obtained a total of 6-8 ground samples for DNA extraction. Among these samples, three samples were used as technical repeats. For soil samples, 0.5 g of soil samples were transferred to 6-8 tubes for preparing DNA extraction. Like plant samples, three samples were used as technical repeats. When sample was failed, DNAs were extracted from remaining transferred samples stored at -80 °C.

30. Were any kind of positive or negative controls used and sequenced? For low-diversity environments and working with many samples it is often needed to check for contaminations.

- Answer: As described in the lines 813-814, we used negative PCR controls that contain not any DNA templates but sterile distilled water during the amplicon generation. When any bands were detected in the negative controls, generated amplicons in the experimental samples were discarded and we repeated the amplicon generation until there were no visible bands in negative controls in an agarose gel. Negative controls that did not show any bands were also used for sequencing to exclude contaminations at the sequence level (Lines 846-847).

31. The statistical tests seem to be done to a high standard and are explained clearly. Some more information on factors included in the analysis (i.e. models) could be provided.

- Answer: More information for analyses was added in the Method section (Lines 859-862, 880).

Reviewer #2 (Remarks to the Author):

In this work, the authors analyzed the bacterial and fungal microbiota associated with rice plants. The experimental design included several plant- and soil compartments (from bulk soil to seeds), across two plant generations (2017-2018) and three sites.

The exploration of the data lead to several observations and conclusions, the most important of which are that seed microbiome included a low number of transmitted OTUs, but these OTUs were among the most abundant ones, and that the assembly of the seed microbiome is affected by the interaction between the plant compartment and the environmental contingencies. Using up-to-date bioinformatic tools, the authors were able to define the ecological mechanisms governing spatio-temporal dynamics

leading to the rice seed microbiome assembly.

I identified the following issues to be solved:

1. - L. 27: mechanisms

- Answer: The word was revised as you commented (Line 28).

2. - L. 85-86: Microbial networks consist of nodes and edges, not the opposite

- Answer: We deleted the sentence (Lines 92-93) following the comment of Reviewer 1 (Comment: Line 85-86: quite a basic sentence that is not needed here).

3. - L. 107: I don't think that the title of this paragraph reflects well its content. In fact, not only temporal changes are described, but also spatial changes (plant compartments)

- Answer: As you commented, we changed the title from "Temporal shift in rice-associated bacterial and fungal communities" to "Temporal and spatial shifts in rice-associated bacterial and fungal communities" (Line 142).

4. - L. 123: years

- Answer: The paragraph where the word was included was revised following the comment of Reviewer 1. As you commented, we carefully checked the sentence where the word "years" was included when revised the paragraph (Line 154).

5. - Figure 1: actually, Alpha-, Gamma- and Deltaproteobacteria are not phyla but classes

- Answer: The figure legend of Fig. 1 was revised to clarify the taxonomic affiliation of Alpha-, Gamma- and Deltaproteobacteria as you commented. You can find the correction in the revised Fig. 1 embedded in the revised manuscript (Line 216) or attached below.

6. - Supplementary Fig. 7: the information that the dissimilarity values relate to the SO seeds (as indicated in the text) should be indicated also in this figure legend

- Answer: The figure caption of Supplementary Fig. 7 was revised by adding the information on the dissimilarity values as you commented. Please see the revised Supplementary Information.

7. - Fig. 5d and Suppl. Fig. 12: the legend states "Connections between taxa are colored by the taxon acting as a source". However, I don't understand how a fungus can be considered as a source of bacteria and vice-versa. Please, clarify.

- Answer: In the network analysis using the Gephi software, two nodes connected by an edge are designated as "Source" with which an edge starts and "Target" with which an edge ends to show the direction of an edge. We think that this term made you confused. To reduce the confusion, we revised the word "source" to "source node" and added additional information on source and target nodes in the captions of Fig. 5d (Lines 520-523) and Supplementary Fig. 12 (in the revised Supplementary Information).

8. - L. 476: did you intentionally exclude seeds from the list of plant compartments, here?

- Answer: We revised the paragraph to which the sentence where you commented belonged following the Reviewer 1's comment. Thanks to your comment, we found another list that we missed seeds in the list of plant compartments. The word "seeds" was added in the list of plant compartments (Line 104).

9. - L. 685: Since you wrote a separate paragraph for DNA extraction, you should delete all the related sentences in the previous paragraph

- Answer: As you commented, we checked and deleted repeated information on sample preparation and DNA extractions in the Method section (Lines 735, 780-783).

10. - L. 834-837: delete these sentences.

- Answer: The sentences were deleted as you commented (Lines 943-946).

REVIEWERS' COMMENTS:

Reviewer #1 (Remarks to the Author):

I am happy with the corrections made by the authors and can now recommend for this article to be published. Great that the patterns stayed (as expected) and now the statistical analyses are fully correct! Thank you for making the corrections.

Point-by-point response to reviewers' comments

Reviewers' comments:

Reviewer #1 (Remarks to the Author):

1. The article has substantially improved and authors have politely and thoroughly answered all my and other reviewers' comments. I only have a minor statistical question on handling the technical replicates which became apparent only after revisions: were the biological and technical replicates treated equally in the statistical models (i.e. PERMANOVA and networks?). Usually one would only include the information of biological replicates in the models as technical replicates are pseudoreplicates and obscure the patterns. The authors should check this and run the relevant models with only using biological replicates to check that the pattern detected is robust enough and biologically relevant. As variation is not shown in most figures, it should not affect the figures too much. If the technical replicates are already omitted or averaged, this is ok and you can ignore this comment.

- Answer: First of all, thanks for your comprehensive comments and suggestions on overall contents of our manuscript, in particular statistical analyses. We checked whether statistical analyses include technical replicates or not. Technical replicates were included in leaves, stems, roots, and soil samples collected in 2018 (Seed samples include only biological replicates; Lines 639-667). Therefore, we checked what you commented in statistical analyses using those samples. As a result, few analyses, including PERMANOVA and PCoA plots in the Fig. 1, and NMDS plots in the Supplementary Fig. 6, were redone, since technical replicates were included. The reanalyzed results for PERMANOVA were described in the Results section (Lines 160-163, 182-187) and the Excel sheet named "PERMANOVA" in the revised Supplementary Data 1. You can also find the revised Fig. 1 and Supplementary Fig. 6 from the line 166 in the revised manuscript and the revised Supplementary Information, respectively, or the figures attached below.

Revised Fig. 1

The PCoA plots in the Fig. 1b and c were revised. Each dot represents each biological replicate of plant and soil compartments.

Revised Supplementary Fig. 6

The NMDS plots were revised. Each dot represents each biological replicate of plant and soil compartments as described in the revised Fig. 1.

I have few tiny corrections further

2. Line 62: maybe add 'open' before or 'remaining' after 'questions'

- Answer: We added the word "open" before the word "questions" as you suggested (Line 56).

3. Line 66: 'knowledge'

- Answer: We corrected the word (Line 59).

Hope my comments were useful, this is really a nice piece of work that is at the same time presented in a very clear and beautiful manner.

Reviewer #2 (Remarks to the Author):

1. The authors satisfactorily addressed my issues.

- Answer: Thanks for your critical comments and delicate suggestions for improving our manuscript.